# LAYOUTNUWA: REVEALING THE HIDDEN LAYOUT EXPERTISE OF LARGE LANGUAGE MODELS

**Zecheng Tang**[1,2*] **Chenfei Wu**[2] **Juntao Li**[1] **Nan Duan**[2†]

[1]Soochow University  [2]Microsoft Research Asia

{zctang@stu., ljt}@suda.edu.cn, {chewu,nanduan}@microsoft.com

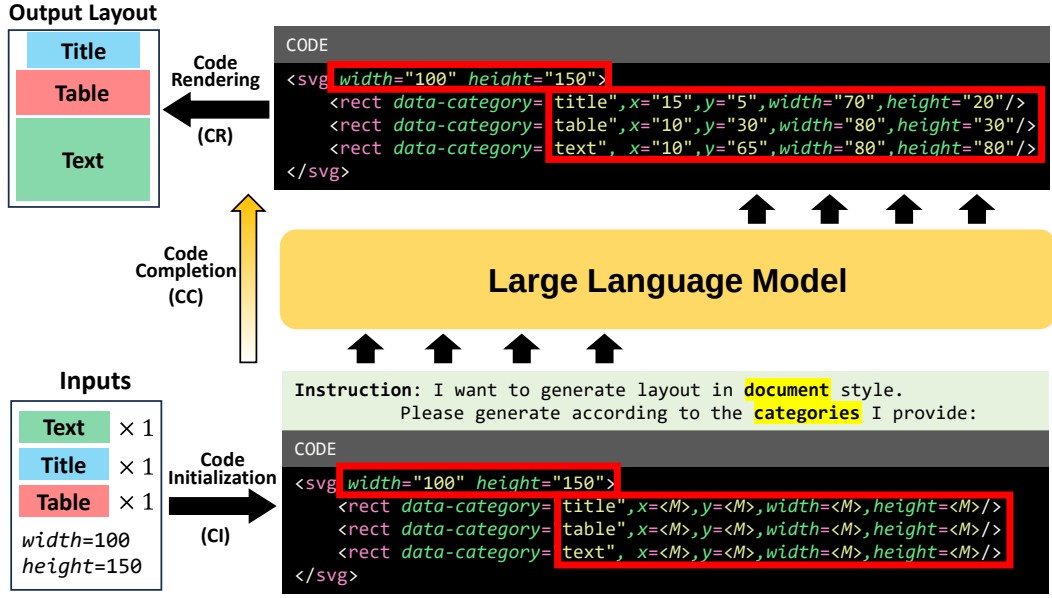

Figure 1: Overview of LayoutNUWA. We propose a Code Instruct Tuning (CIT) approach that consists of three modules: 1) the Code Initialization (CI) module quantifies the numerical conditions and initializes them as an HTML code with masks; 2) the Code Completion (CC) module utilizes the knowledge of large language models to complete the masked portions within the HTML code; 3) the Code Rendering (CR) module directly renders the completed code into the final graphic layout.

## ABSTRACT

Graphic layout generation plays a significant role in user engagement and information perception. Existing methods primarily treat layout generation as a numerical optimization task, focusing on quantitative aspects while overlooking the semantic information of layout, such as the relationship between each layout element. In this paper, we propose LayoutNUWA, the first model that treats layout generation as a code generation task to enhance semantic information and harnesses the hidden layout expertise of large language models (LLMs). More concretely, we develop a Code Instruct Tuning (CIT) approach comprising three interconnected modules: 1) the Code Initialization (CI) module quantifies the numerical conditions and initializes them as HTML code with strategically placed masks; 2) the Code Completion (CC) module employs the formatting knowledge of LLMs to fill in the masked portions within the HTML code; 3) the Code Rendering (CR) module transforms the completed code into the final layout output, ensuring a highly interpretable and transparent layout generation procedure that directly maps code to a visualized layout. We attain significant state-of-the-art performance (even over 50% improvements) on multiple datasets, showcasing the strong capabilities of LayoutNUWA. Our code is available at https://github.com/ProjectNUWA/LayoutNUWA.

---

[*]Both authors contributed equally to this research. During Zecheng's internship under the mentorship of Chenfei at MSRA.

[†]Corresponding author.

# 1 INTRODUCTION

Graphic layout, which refers to the organization and positioning of design elements, significantly influences the way users engage with and perceive the presented information (Lee et al., 2020). As a growing research field, layout generation (Li et al., 2019; Yang et al., 2020) aims to create diverse and realistic layouts that streamline the design process and cater to various applications, such as user interfaces (Deka et al., 2017; Jiang et al., 2022), indoor scenes (Di & Yu, 2021; Feng et al., 2023), document layouts  (Zheng et al., 2019; Yamaguchi, 2021), presentation slides (Fu et al., 2022), etc.

Current approaches (Jyothi et al., 2019; Li et al., 2019; Arroyo et al., 2021; Zhang et al., 2023a) regard each element in the layout as numerical tuples $(c, x, y, w, h)$, in which $c$ indicates the element category, $x$ and $y$ represent coordinates, $w$ and $h$ correspond to width and height. For example, autoregressive-based methods (Yang et al., 2020; Jiang et al., 2022) view the tuple as a sequence and predict their values sequentially, while diffusion-based methods (Chai et al., 2023; Inoue et al., 2023) consider the tuple as a whole and predict their values through a denoising approach. Despite adopting different generative models, all of these methods fundamentally consider layout generation as a numerical tuple optimization task. However, representing layouts as numerical tuples has its limitations, as it primarily focuses on capturing the quantitative aspects of the layout, such as positions and sizes, while lacking semantic information, e.g., the attribute of each numerical value, which may limit the model's ability to capture more complex and rich layout information.

An insightful question emerges from the limitations of existing methods in layout generation: can we integrate semantic information into the layout generation process to enrich the overall representation and enhance the quality of the generated layouts? Addressing this question brings forth two major benefits: firstly, it bolsters the understanding of relationships among various layout elements, and secondly, it enables us to tap into the semantic capabilities of LLMs (Tang et al., 2023), resulting in more intricate and contextually relevant layouts for a wide range of applications (Jiang et al., 2022). Considering the inherent logical nature of layouts, which involve dependency relationships among layout elements, and the fact that each graphic layout can be represented with a fixed structure sequence, code languages emerge as a promising alternative. Code languages can encompass numerical and semantic information while possessing a strong logical foundation (Chen et al., 2022), which can thus bridge the gap between existing methods and the desired enriched representation.

Based on the above observations, we propose LayoutNUWA, a groundbreaking model that revolutionizes the layout generation task by treating it as a code generation task. Our innovative approach is designed to not only enhance the semantic information within layouts but also seamlessly leverage the expertise of LLMs in the layout generation process. To achieve this, we design a Code Instruct Tuning (CIT) approach comprising three interconnected modules: 1) firstly, the Code Initialization (CI) module quantifies the numerical conditions and initializes them as HTML code with strategically placed masks, paving the way for more meaningful and coherent layouts; 2) secondly, the Code Completion (CC) module employs the formatting knowledge of LLMs to fill in the masked portions within the HTML code, thereby harnessing the power of LLMs to improve the accuracy and consistency of the generated layouts; 3) lastly, the Code Rendering (CR) module transforms the completed code into the final layout output, ensuring a highly interpretable and transparent layout generation procedure that directly maps code to a visualized layout.

Experiments across a variety of conditional layout generation tasks on three datasets, i.e., Rico (Deka et al., 2017), PubLayNet (Zhong et al., 2019) and Magazine (Zheng et al., 2019), highlight the superiority of our method, in which LayoutNUWA can significantly outperform all the baselines and shows comparable results with the task-specific models. Furthermore, LayoutNUWA can achieve at least a 50% improvement in performance compared to the best baseline on the low-resource datasets, e.g., the Magazine dataset. In a nutshell, our contributions can be outlined as follows:

- We introduce LayoutNUWA, the first model that treats the layout generation task as a code generation task, effectively harnessing the hidden layout expertise of LLMs.

- We propose Code Instruct Tuning, which empowers the model to adhere to instructions and enriches the semantic information of layout, resulting in precise and standardized code.

- We attain significant state-of-the-art performance on multiple datasets, showcasing the robust capabilities of LayoutNUWA.

## 2 RELATED WORK

### 2.1 LAYOUT GENERATION

Automatic layout generation, an important task for automatic graphical design for various scenarios such as document layouts (Zheng et al., 2019; Zhong et al., 2019; Yamaguchi, 2021; Fu et al., 2022), posters (Yang et al., 2016; Guo et al., 2021; Li et al., 2023) and user interface (Deka et al., 2017), has been recently extensively researched. Early approaches for layout generation involve embedding design rules into manually-defined energy functions (O'Donovan et al., 2014; O'Donovan et al., 2015), while other methods have explored generative models such as GANs and VAEs for generating numerical graphic and scene layouts, including LayoutGAN (Li et al., 2019), LayoutVAE (Jyothi et al., 2019), LayoutGAN++ (Kikuchi et al., 2021), NDN (Lee et al., 2020) and READ (Patil et al., 2020). Apart from them, transformer-based approaches utilize self-attention mechanisms to learn numerical contextual relationships between elements and achieve layout completion based on partial layout inputs (Yang et al., 2020; Kong et al., 2022; Feng et al., 2023). Recently, with the prevalence of diffusion models, several works also adopted diffusion models to tackle a broader range of conditional layout generation (Chai et al., 2023; Inoue et al., 2023; Zhang et al., 2023a; Hui et al., 2023; Cheng et al., 2023). However, existing methods primarily treat layout generation as a numerical optimization task, focusing on quantitative aspects while overlooking the semantic information of layout, such as the relationship between each layout element. Different from previous works, we convert the layout generation task into the code generation task to directly generate the layout in code language and thus utilize the rich knowledge from LLMs, which can significantly improve the FID by 50% in the Magazine dataset in § 4.2.

### 2.2 INSTRUCTION TUNING

Instruction tuning represents the process of fine-tuning LLMs on the instruction dataset in a supervised fashion, which narrows the gap between the next-word prediction manner of LLMs and the users' objective of having LLMs adhere to human instructions (Zhang et al., 2023c). Early attempts on instruction tuning involve multi-task training with manually-written descriptions about different tasks (Mishra et al., 2021; Wei et al., 2021; Sanh et al., 2021; Xu et al., 2022; Muennighoff et al., 2022; Iyer et al., 2022) or automatically generated instructions (Wang et al., 2022; Gu et al., 2022; Zhang et al., 2023b; Honovich et al., 2022a;b). Apart from controlling the LLMs through input instruction, Nye et al. (2021) show that LLM can handle more complex tasks by generating the intermediate steps and Wei et al. (2022) propose chain-of-thought technique by enriching the instruction with intermediate reasoning step descriptions, which endows LLMs with better performance (Wang et al., 2022; Zelikman et al., 2022; Wu et al., 2023; Xu et al., 2023). However, the instruction tuning methods mentioned above are primarily intended for text generation tasks and not ideal for layout generation tasks, which involve numerical optimization. Thus, we propose a code instruction tuning method that is specially designed for the layout generation task. Experiments in § 5.1 indicate that the performance significantly drops if the code instruction tuning is not adopted.

## 3 METHODOLOGY

### 3.1 PROBLEM FORMULATION

The layout generation task aims to generate a well-organized layout $\mathcal{S} = \{s_i\}_{i=1}^N$, with $N$ representing the number of elements in the layout. Each element, $s_i = (c_i, x_i, y_i, w_i, h_i)$, consists of the following components: $c_i$ is the category, $x_i, y_i$ indicate the center location, and $w_i, h_i$ represent the width and height, respectively. In this study, we focus on the conditional layout generation task, wherein partial components in $s_i$ are masked with $M$, and the complete layout $S$ should be predicted by model $f_\theta$ conditioned on the remaining components $S_{\setminus M}$:

$$\mathcal{S} = f_\theta(\mathcal{S}_{\setminus M}) \tag{1}$$

Previous works (Jyothi et al., 2019; Yang et al., 2020; Inoue et al., 2023) regard each element $s_i$ as a sequence of numerical values, e.g., (0, 10, 20, 25, 30), and train a model to directly generate these values. However, this approach overlooks the semantic information of the components, thus limiting the model's understanding of the layout semantics. Based on this observation, we propose

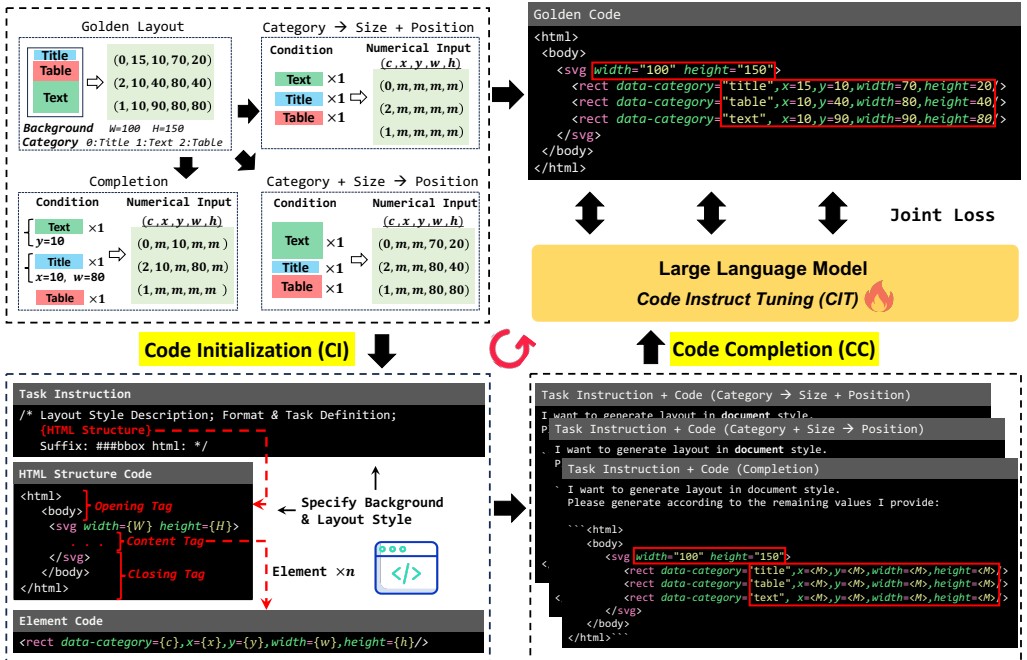

Figure 2: The training process of LayoutNUWA, which converts layout generation task to code generation task and utilizes a code instruct tuning to leverage LLM's capability for layout generation.

a new problem definition, where we convert the input $S_{\setminus M}$ and output $S$ into a code language and view the layout generation task as a code generation task:

$$\text{CODE}(\mathcal{S}) = f_\theta(\text{CODE}(\mathcal{S}_{\setminus M})) \tag{2}$$

Eq. 2 has the following 3 advantages compared with Eq. 1:

- **Semantic Insights**: By converting the numerical values into code language, the model can better capture the semantic relationships between different components of the layout.

- **LLM Utilization**: By using code language, the model can further leverage the knowledge of Large Language Models (LLMs) and thus enhance the quality of the generated layouts.

- **Model Scalability**: The code language has a stronger expressive capability compared to numerical values, which allows the addition of more attributes for layout elements.

## 3.2 CODE INSTRUCT TUNING

As shown in Fig. 1, we propose Code Instruct Tuning (CIT) with three modules: (1) *Code Initialization* module converts layout into masked code language with dynamic templates; (2) *Code Completion* module inputs the masked code to LLMs to generate complete code; (3) *Code Rendering* module directly renders code to the final graphic layout. We illustrate these modules below.

### 3.2.1 CODE INITIALIZATION

**Element Quantization** We quantify the numerical values of $i$-th element position $\{x_i, y_i\}$ and size $\{w_i, h_i\}$ in the layout with Adaptive Quantization method (Inoue et al., 2023) that applies $k$-Means algorithm (MacQueen et al., 1967) to cluster the position and size information of each element, addressing the highly imbalanced distribution of these values, e.g., elements may overlap or cluster together. Different from the previous works (Chai et al., 2023; Zhang et al., 2023a; Inoue et al., 2023), we use absolute position to represent the coordinates rather than relative positions. This aligns with code language and allows direct rendering of layouts without necessitating coordinate conversion, thereby preventing potential information loss. We maintain precision up to one decimal place and directly convert the clustered results into strings.

**Template Construction**    The overview of template construction is shown in Fig. 2. We construct the templates based on the most common web page layout code, HTML, which contains a wealth of information and is easily accessed by LLMs during the pre-training process (Touvron et al., 2023; Rozière et al., 2023). Specifically, in HTML code, each element is described with a tag that provides information about the content or the element structure. Since the elements in the layout are regular squares, we chose the `<rect>` tag as the content tag to describe each element:

```
<rect data-category={c_i} x={x_i} y={y_i} width={w_i} height={h_i}>
```

where $c_i$ is the element category in textual format and $\{x_i, y_i, w_i, h_i\}$ are the quantified position and size of the $i$-th element. Then, to combine all the elements into a unified structure, we used an opening tag and a closing tag to define the boundaries of each layout, which can be written as:

```
<html><body><svg width={W} height={H}> ... </svg></body></html>
```

where $W$ and $H$ are the background width and height of the layout.

In order to facilitate better learning of layout in various domains and tasks and leverage the instruction-following capabilities of LLMs, we design the following prompts:

```
I want to generate layout in {Domain} style. Please generate the
   layout according to the {Task Condition} I provide:
```

where the {`domain`} and the {`Task Condition`} will vary according to different domains and tasks. For instance, for the RICO dataset, we set `Domain` as "mobile UI", and for the layout completion task, we set `Task Condition` as "remaining values".

### 3.2.2    CODE COMPLETION

To construct the conditional input of the layout generation task, we utilize the mask tokens of LLMs to represent the masked values $M$ and let the model predict the masked values within the HTML code. Different from previous works (Chai et al., 2023; Zhang et al., 2023a; Inoue et al., 2023) that applied the customized numerical vocabulary, we employ the LLM's token vocabulary directly. By doing so, we can leverage the knowledge of the numerical tokens inherited in the LLMs. Considering that almost all the LLMs follow auto-regressive generation manner that brings significant limitation to the layout generation task since the model should predict the same layout under different element orders, even if the layout doesn't have a naturally defined order (Yang et al., 2020). Thus, we design a self-consistency strategy that randomly permutes the order of the input elements in the layout within a mini-batch. Meanwhile, in order to adapt LLMs to different conditional layout generation tasks, we have performed multi-task modeling on the same layout, utilizing various conditions and implementing a joint loss for these tasks. Given the permutation times $K$ and task numbers $T$, the joint loss, denoted as $L(\cdot)$, for each layout $\mathcal{S}$ can be written as:

$$L(\mathcal{S} \mid \theta) = \sum_{t=1}^{T} \sum_{j=1}^{N} \sum_{k=1}^{K} L(s_j^{(k)} \backslash M_j^{(t)} \mid \theta),  \tag{3}$$

where $\theta$ is the model parameters and $s_j$ denote the $j$-th element in the layout $\mathcal{S}$.

### 3.2.3    CODE RENDERING

Most existing works require the extra conversion step to render the graphic layouts (Yang et al., 2020; Chai et al., 2023; Zhang et al., 2023a), e.g., converting the relative position to the absolute position, causing the information loss. Different from previous work, LayoutNUWA allows for immediate rendering as it generates the absolute position directly. Besides, considering the potential output issues such as boundary overflow (Inoue et al., 2023) and format errors, we employ regular expressions to remove mismatched formats and implement clipping operations for elements that exceed the background size.

## 4 EXPERIMENT

### 4.1 EXPERIMENTAL SETTINGS

**Datasets**   We evaluate the model performance on three widely used public datasets. RICO (Deka et al., 2017) is a user interface design dataset for mobile applications containing 25 element categories and 66K+ UI layouts.  PubLayNet (Zhong et al., 2019) consists of 360K+ layouts for documents with 5 element categories. Magazine (Zheng et al., 2019) is a low-resource magazine layout dataset containing around 4K annotated layouts and 6 element categories. We follow LayoutDM (Inoue et al., 2023) to view the original validation data as the testing set and pre-process all three datasets by discarding the layouts containing more than 25 elements as well as splitting the filtered data into the training and new validation sets by 95% and 5%.

**Evaluation Metrics**   We employ four metrics to evaluate the generation results comprehensively, including Frechet Inception Distance (FID), Maximum Interaction over Union (mIoU), Alignment (Align.), and Overlap.  Among them, FID compares the distribution of generated and real layouts. Similar to the previous work (Inoue et al., 2023), we utilize an enhanced feature extraction model for layouts (Kikuchi et al., 2021) to compute the FID score. We measure the conditional similarity between generated and real layouts using mIoU, which is done by calculating the maximum IoU between bounding boxes of generated and real layouts with the same type set. Alignment and Overlap scores are calculated following the previous work (Li et al., 2019) to evaluate proper element alignment and overlapping in a generated layout, and it is worth noting that we ignore normal overlaps, e.g., elements on top of the background, and discard the layouts that failed to generate.  For reference, we show the evaluation results between the validation set and test set as Real data.

**Tasks and Baselines**   We evaluate LayoutNUWA on three conditional layout generation tasks[1], including the Category to Size and Position (C $\rightarrow$ S+P) task, the Category and Size to Position (C+S $\rightarrow$ P) task, and the Completion task.  More concretely, the C $\rightarrow$ S+P task requires the model to predict the position and size of the element based on its category. For the C+S $\rightarrow$ P task, the model predicts the position of the element based on both its size and category.  Finally, in the completion task, the element's size and position values are randomly masked up to 80%, and the model predicts the entire layout using the remaining values. We compare LayoutNUWA with six strong baselines, including LayoutTrans (Yang et al., 2020), BLT (Kong et al., 2022), LayoutGAN++ (Kikuchi et al., 2021), MaskGIT (Chang et al., 2022), DiffusionLM (Li et al., 2022) and LayoutDM (Inoue et al., 2023). For the above baselines, we implement them with the official code directly.

**Implementation Details**   We implement LayoutNUWA with two 7B LLMs: LLaMA2 (L2) (Touvron et al., 2023) and CodeLLaMA (CL) (Rozière et al., 2023).  We train LayoutNUWA with two settings: (1) Domain-Specific (DS) setting, where the model is trained on distinct datasets, and (2) Domain-Agnostic (DA) setting, where the model is trained on all three datasets, including RICO, PubLayNet, and Magazine. The default configuration for LayoutNUWA utilizes CodeLLaMA (CL) and Domain-Agnostic (DA), i.e., LayoutNUWA-CL-DA. We set permutation times $K = 10$ and task numbers $T = 3$. For model training, we use DeepSpeed Library (Rajbhandari et al., 2020) to run all experiments on 64 NVIDIA V100 GPUs. We apply Top-$p$ sampling (Holtzman et al., 2019) for inference, where $p = 0.9$ and the temperature is 0.6. We set the maximum generation length as 1024 for each sample to ensure the completeness of the layout code.

### 4.2 QUANTITATIVE EVALUATION

We report the model performance on three datasets: the Magazine dataset in Tab. 1, RICO, and PubLayNet datasets in Tab. 2. For the Magazine dataset, LayoutNUWA demonstrates a remarkable performance by significantly surpassing all baseline measures across all tasks. Moreover, it outperforms the strong baseline LayoutDM by more than 50% when assessed with the FID metric.

The significant improvements in Tab. 1 are due to three aspects: 1) previous approaches generated numerical values, while LayoutNUWA generates code with labels, which greatly benefits the model

---

[1]We also report the model performance on three datasets under the unconditional generation setting in Appendix E for space limitation.

| Model | Layout Format | LLM | Domain | C → S + P | | C + S → P | | Completion | |
|---|---|---|---|---|---|---|---|---|---|
| | | | | mIOU (↑) | FID (↓) | mIOU (↑) | FID (↓) | mIOU (↑) | FID (↓) |
| LayoutTrans | Numerical | - | Specific | 0.116 | 36.207 | 0.153 | 33.931 | 0.228 | 25.804 |
| BLT | Numerical | - | Specific | 0.087 | 65.372 | 0.126 | 41.089 | 0.103 | 97.142 |
| LayoutGAN++ | Numerical | - | Specific | 0.259 | 16.952 | 0.293 | 11.569 | - | - |
| MaskGIT | Numerical | - | Specific | 0.059 | 140.94 | 0.100 | 78.226 | 0.024 | 152.591 |
| DiffusionLM | Numerical | - | Specific | 0.151 | 32.114 | 0.144 | 24.370 | 0.138 | 33.172 |
| LayoutDM | Numerical | - | Specific | 0.234 | 19.206 | 0.308 | 14.265 | 0.328 | 15.804 |
| LayoutNUWA-L2-DS (ours) | Code | LLaMA2 | Specific | 0.260 | 9.741 | 0.358 | _6.682_ | _0.418_ | 8.257 |
| LayoutNUWA-L2-DA (ours) | Code | LLaMA2 | Agnostic | _0.293_ | 9.632 | _0.394_ | 7.238 | 0.413 | 8.734 |
| LayoutNUWA-CL-DS (ours) | Code | CodeLLaMA | Specific | 0.293 | _8.985_ | 0.348 | **5.355** | 0.410 | **7.341** |
| LayoutNUWA (ours) | Code | CodeLLaMA | Agnostic | **0.312** | **8.791** | **0.418** | 6.755 | **0.495** | _7.572_ |
| **Real Data** | - | - | - | 0.348 | 6.695 | 0.348 | 6.695 | 0.348 | 6.695 |

Table 1: Quantitative comparison on Magazine dataset, where the bold font denotes the best result and underline represents the second-best performance.

| Tasks | Models | RICO | | | | PubLayNet | | | |
|---|---|---|---|---|---|---|---|---|---|
| | | mIoU (↑) | Align. (→) | Overlap (→) | FID (↓) | mIoU (↑) | Align. (→) | Overlap (→) | FID (↓) |
| Condition C → S + P | LayoutTrans | 0.219 | 0.014 | 13.012 | 11.237 | 0.271 | 0.016 | 3.229 | 38.910 |
| | BLT | 0.203 | 0.013 | 11.743 | 14.260 | 0.232 | 0.009 | 16.742 | 76.499 |
| | LayoutGAN++ | 0.263 | 0.016 | 3.544 | 6.842 | 0.354 | 0.011 | 1.713 | 10.219 |
| | MaskGIT | 0.267 | 0.001 | 26.865 | 27.470 | 0.320 | 0.004 | 1.857 | 16.898 |
| | DiffusionLM | 0.299 | 0.018 | 17.655 | 31.644 | 0.262 | 0.027 | 3.532 | 20.021 |
| | LayoutDM | 0.275 | 0.010 | 11.938 | 3.576 | 0.310 | 0.010 | 0.024 | 7.915 |
| | LayoutNUWA-L2-DS (ours) | 0.351 | _0.009_ | _10.190_ | 3.728 | 0.337 | 0.009 | _0.058_ | 6.986 |
| | LayoutNUWA-L2-DA (ours) | _0.386_ | 0.011 | 10.214 | _3.101_ | 0.324 | 0.011 | 0.077 | 6.890 |
| | LayoutNUWA-CL-DS (ours) | 0.377 | 0.009 | 10.263 | 3.706 | _0.376_ | _0.008_ | 0.053 | _6.715_ |
| | LayoutNUWA (ours) | **0.445** | **0.004** | **7.943** | **2.524** | **0.385** | **0.001** | 0.086 | **6.579** |
| Condition C + S → P | LayoutTrans | 0.311 | 0.011 | 11.902 | 9.368 | 0.315 | 0.013 | 2.531 | 31.627 |
| | BLT | 0.341 | 0.008 | 13.470 | 4.487 | 0.356 | _0.006_ | 5.469 | 8.831 |
| | LayoutGAN++ | 0.349 | 0.011 | _9.628_ | 6.219 | 0.346 | 0.008 | 2.746 | 9.936 |
| | MaskGIT | 0.331 | **0.003** | 26.390 | 12.898 | 0.384 | 0.005 | 1.950 | 5.453 |
| | DiffusionLM | 0.278 | 0.020 | 11.884 | 15.931 | 0.324 | 0.014 | 3.990 | 16.407 |
| | LayoutDM | 0.391 | 0.009 | 12.072 | **2.288** | 0.381 | 0.010 | 2.041 | 4.175 |
| | LayoutNUWA-L2-DS (ours) | 0.462 | 0.008 | 10.436 | 3.035 | 0.426 | 0.010 | 1.752 | 4.105 |
| | LayoutNUWA-L2-DA (ours) | 0.464 | _0.007_ | 10.117 | 2.973 | 0.464 | 0.009 | 1.984 | _3.993_ |
| | LayoutNUWA-CL-DS (ours) | 0.469 | _0.007_ | 9.856 | 2.984 | _0.466_ | 0.009 | _1.610_ | 4.012 |
| | LayoutNUWA (ours) | **0.564** | _0.007_ | **7.968** | _2.870_ | **0.483** | **0.002** | 0.108 | **3.697** |
| Completion | LayoutTrans | 0.561 | 0.008 | 10.080 | 3.733 | 0.439 | 0.012 | 2.053 | 8.689 |
| | BLT[†] | 0.471 | **0.007** | 53.658 | 121.110 | 0.157 | **0.002** | 109.483 | 155.157 |
| | MaskGIT | 0.537 | 0.024 | 9.242 | 33.463 | 0.349 | 0.011 | 4.768 | 12.013 |
| | DiffusionLM | 0.218 | 0.021 | **8.681** | 22.220 | 0.332 | 0.012 | 4.436 | 16.576 |
| | LayoutDM | 0.580 | _0.009_ | 15.676 | 9.224 | 0.377 | 0.011 | 1.891 | 7.570 |
| | LayoutNUWA-L2-DS (ours) | 0.610 | 0.009 | 7.239 | 8.875 | 0.407 | 0.010 | 1.337 | 7.337 |
| | LayoutNUWA-L2-DA (ours) | _0.624_ | **0.007** | 10.457 | _8.724_ | _0.477_ | 0.012 | 1.383 | _7.169_ |
| | LayoutNUWA-CL-DS (ours) | **0.641** | **0.007** | 7.529 | 8.734 | 0.473 | 0.012 | _1.311_ | 7.233 |
| | LayoutNUWA (ours) | 0.616 | **0.007** | _8.123_ | **7.542** | **0.481** | _0.009_ | **1.292** | **6.929** |
| **Real Data** | - | 0.438 | 0.004 | 8.706 | 6.250 | 0.691 | 0.001 | 0.039 | 1.850 |

Table 2: Quantitative comparison on the RICO and PubLayNet Datasets. For Align. and Overlap metrics, the closer to the real data, the better performance is (indicated by →).

by utilizing the semantic information of layout attributes such as width, height, position, and category; 2) none of the previous methods used LLMs. However, we have introduced LLMs for the first time, which has resulted in significant performance enhancements, i.e., performance has improved from 19.206 to 9.741. Furthermore, when we use CodeLLaMA, which is tuned on code language, the performance improves even further to 8.985; 3) since different domains require distinct layout formats, early numerical-based methods could only be trained in a domain-specific manner. However, LayoutNUWA is based on code structure, which can be trained in a domain-agnostic manner, allowing for complementary among data from various domains, thus further improving FID to 8.791.

We have also conducted extensive experiments on two other datasets: RICO and PubLayNet, as shown in Tab. 2. The LayoutNUWA notably surpasses all baseline methods in the majority of tasks. Although it does not achieve the best performance in two specific tasks, it still secures at least the second-highest performance in those instances. This shows the strong generalization of the LayoutNUWA. It is worth mentioning that our model also achieves closer Align. and Overlap scores to the Real Data compared to the baselines. Although previous work has suggested that refinement and discriminator processes can contribute to improving the Align. and Overlap (Inoue et al., 2023; Li et al., 2019) scores, our method attains better results without employing these steps.

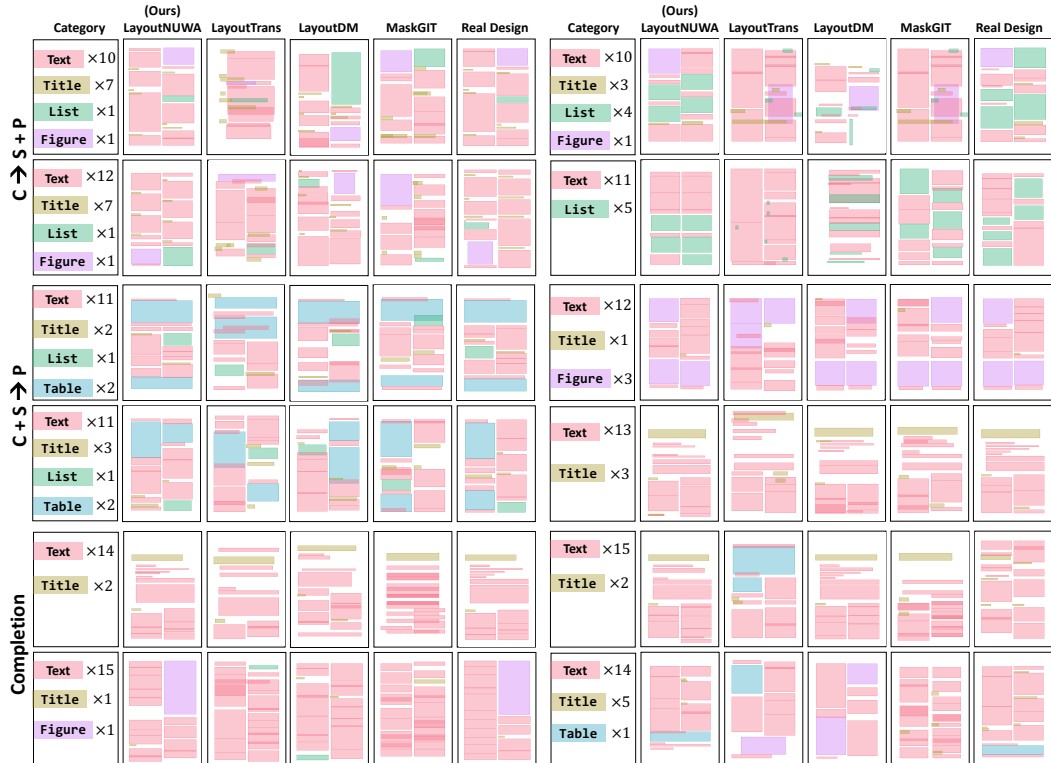

Figure 3: Samples generated by LayoutNUWA on the PubLayNet dataset.

| Task | Models | Tuning Method | mIoU (↑) | Align. (→) | Overlap (→) | FID (↓) | Fail (↓) |
|------|--------|---------------|----------|------------|-------------|---------|----------|
| **Condition C → S + P** | LayoutNUWA-L2-DS | CTT | **0.260** | **0.021** | **2.898** | **9.741** | **0.000 %** |
| | w/o template | Instruct Tuning (DS) | 0.124 | 0.049 | 3.221 | 16.324 | 1.020 % |
| | w/o template | Instruct Tuning (DA) | - | - | - | - | 0.000 % |
| | w/o template&instruct | Numerical Tuning | 0.126 | 0.053 | 3.581 | 17.982 | 3.571 % |
| **Condition C + S → P** | LayoutNUWA-L2-DS | CIT | **0.358** | **0.020** | **2.483** | **4.682** | **0.000 %** |
| | w/o template | Instruct Tuning (DS) | 0.182 | 0.021 | 2.673 | 12.432 | 0.000 % |
| | w/o template | Instruct Tuning (DA) | - | - | - | - | 0.000 % |
| | w/o template&instruct | Numerical Tuning | 0.189 | 0.024 | 2.892 | 14.326 | 0.000 % |
| **Completion** | LayoutNUWA-L2-DS | CIT | **0.418** | **0.020** | **2.309** | **7.257** | **0.253 %** |
| | w/o template | Instruct Tuning (DS) | 0.206 | 0.017 | 2.882 | 15.732 | 5.102 % |
| | w/o template | Instruct Tuning (DA) | - | - | - | - | 6.633 % |
| | w/o template&instruct | Numerical Tuning | 0.214 | 0.020 | 3.003 | 16.243 | 6.122 % |
| **Real Data** | - | - | 0.348 | 0.016 | 1.521 | 6.695 | - |

Table 3: Comparison among different tuning methods, where "Fail" is the failure ratio of generation.

## 4.3 QUALITATIVE EVALUATION

We render the generated layout code with the Code Rendering (CR) method, and Fig. 3 shows the sampled rendering results of the PubLayNet dataset. By comparing with other baselines, we can observe that the layouts generated by LayoutNUWA exhibit excellent element alignment, and the proportion of overlap between elements is minimal. Additionally, our results are the most consistent with the Real Design data, i.e., the size and position of the generated element are essentially consistent with the real design, indicating that by treating the layout generation task as a code generation task, LayoutNUWA has successfully learned the distribution of document layouts, thus result in more precise and realistic layouts. More generated cases can be referred to Fig. 11 in the appendix.

## 5 ABLATION STUDY

We investigate the effectiveness of the CIT tuning method in Sec. 5.1 and compare the impact of different output formats and fine-tuning in Sec. 5.2. More concretely, we set the LayoutNUWA-L2-DS model as the basic setting and conduct the ablation studies on the Magazine dataset.

| Task | Model | Layout Format | mIoU ($\uparrow$) | Align. ($\rightarrow$) | Overlap ($\rightarrow$) | FID ($\downarrow$) | Fail ($\downarrow$) |
|---|---|---|---|---|---|---|---|
| **Condition** **C $\rightarrow$ S + P** | LayoutNUWA-N | Numerical | 0.000 | 0.000 | 0.867 | - | 78.030 % |
| | LayoutNUWA-L2-DS | Code | **0.260** | **0.021** | **2.898** | **9.741** | **0.000 %** |
| **Condition** **C + S $\rightarrow$ P** | LayoutNUWA-N | Numerical | 0.000 | 0.000 | 24.959 | 349.231 | 21.717 % |
| | LayoutNUWA-L2-DS | Code | **0.358** | **0.020** | **2.483** | **4.682** | **0.000 %** |
| **Completion** | LayoutNUWA-N | Numerical | 0.000 | 0.000 | 16.602 | - | 29.293 % |
| | LayoutNUWA-L2-DS | Code | **0.418** | **0.020** | **2.309** | **7.257** | **0.253 %** |
| **Real Data** | - | - | 0.348 | 0.016 | 1.521 | 6.695 | - |

Table 4: Comparison among different output formats.

## 5.1 EFFECT OF TUNING METHODS

We progressively reduce the modules in CIT and fine-tune the model using the corresponding constructed data. Specifically, we first exclude the code template and directly convert the element information into an ordered sequence $S$ with a task instruction before it, i.e., the instruction tuning method. Then, we further remove the task instruction and directly fine-tune the model using data from different tasks separately, i.e., the numerical tuning method. As shown in Tab. 3, we can observe that the model performance has declined significantly without the code template, and it can only work in the DS setting since the model can simply generate repetitive and out-of-order results that are inconsistent with the element sequence in the DA setting. Furthermore, the numerical tuning method can only support the DS setting as there is no task instruction for the model to distinguish between different tasks, and the model performance is far inferior compared to those of the CIT as such an approach overlooks the rich semantic information among the elements and can not calibrate the prior code knowledge of LLMs.

## 5.2 EFFECT OF OUTPUT FORMAT AND FINETUNING

We compared the effects of the model output in code format and numerical format. For the numerical output format, we designed a Code Infilling task, which involves making the LLM predict only the masked values rather than predicting the entire code sequence. As shown in Tab. 4, we can find that generating in numerical format will increase the failure ratio of model generations, e.g., the model will

| Model | C $\rightarrow$ S + P | C + S $\rightarrow$ P | Completion |
|---|---|---|---|
| | Fail ($\downarrow$) | Fail ($\downarrow$) | Fail ($\downarrow$) |
| LLaMA2 (Zero-Shot) | 100.0 % | 100.0 % | 100.0 % |
| CodeLLaMA (Zero-shot) | 100.0 % | 100.0 % | 100.0 % |
| GPT-4 (Zero-Shot) | 34.2 % | 28.8 % | 28.5 % |
| LayoutNUWA | **0.0 %** | **0.0 %** | **0.3 %** |

Table 5: Comparison with LLMs.

generate repetitive results, and significantly decrease the model performance. This is because the layout generated by the conditional layout generation task should be logical, while only predicting the masked parts can lead to discrete values that lack logic. Besides, Due to the influence of the autoregressive manner, where the content generated in the next step depends on the previous history, this phenomenon may result in a higher failure probability of model generation when predicting layouts with more masked values. We also conduct a comparison between LayoutNUWA and GPT-4 (Bubeck et al., 2023). Specifically, we allow GPT-4 to perform inference by constructing the input using the CIT method. Tab. 5 shows code instruct tuning for LLM is necessary, as using LLM in a zero-shot manner leads to a high fail rate (100% fail rate of LLaMA2 and around 30% for GPT-4).

## 6 CONCLUSION

In this paper, we propose LayoutNUWA, a groundbreaking approach that treats layout generation as a code generation task, effectively enriching the semantic information of layouts and leveraging the hidden expertise of LLMs. Extensive experiments on multiple datasets have demonstrated the superiority of our method. This research has the potential to revolutionize the field of layout generation and pave the way for further exploration and development of semantic-aware layout generation approaches in various applications.

ETHICS STATEMENT

This research is done in alignment with Microsoft's responsible AI principles.

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

# A LIMITATIONS

Since LayoutNUWA employs the autoregressive (AR) LLMs as the backbone, our method naturally inherits the shortcomings of the AR models:

- The generation speed is slower than the non-autoregressive models (Chang et al., 2022).

- It suffers from the error propagation problem (Wu et al., 2018), i.e., as shown in Fig. 4, especially when training is insufficient, where the content generated later in the sequence may be negatively affected by the errors in the content generated earlier.

Additionally, the model may suffer from the domain confusion issue Long et al. (2022) to some extent under the Domain-Agnostic setting as illustrated in Appendix F. In our future work, we will address these challenges and make improvements to generate better graphic layouts.

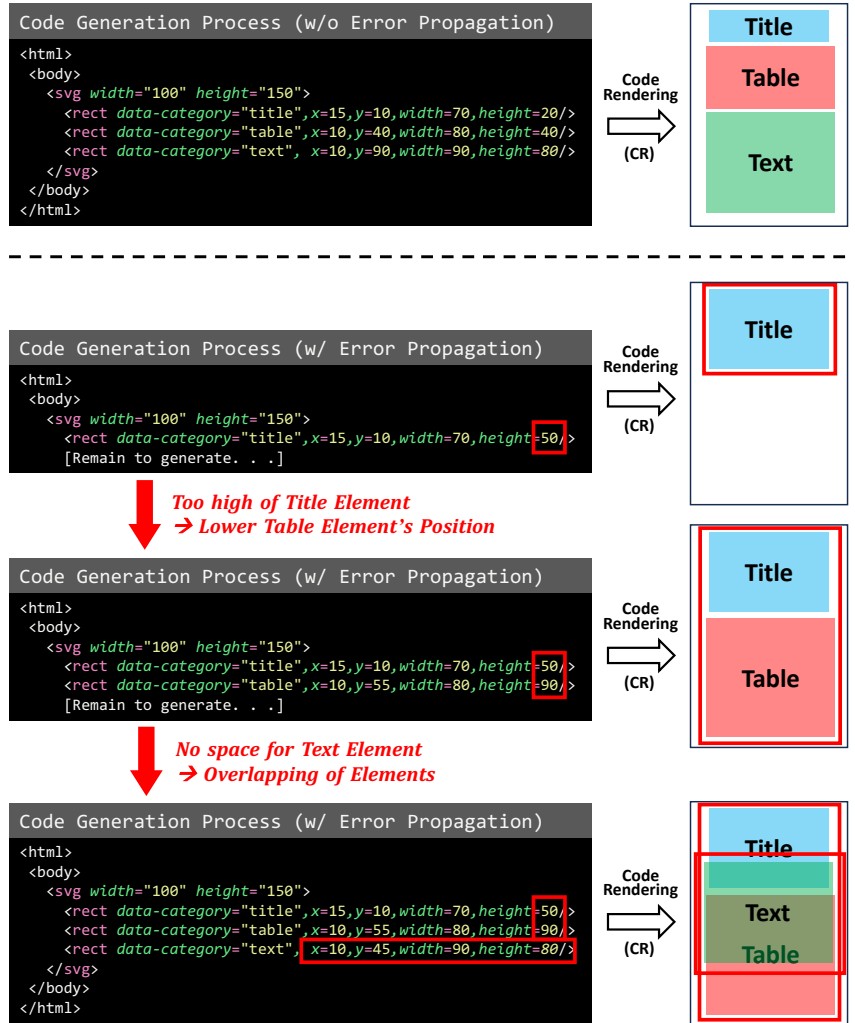

Figure 4: One possible case illustrating error propagation, where the top group shows an ideal model generation scenario, and the bottom group shows the error propagation process.

## B    COMPARISON WITH GPT-4

We utilize the GPT-4 model with the commercial API and strictly follow the usage policy [2]. We report the detailed performance of the GPT-4 model in Tab. 6 and show several rendered graphic layouts in Fig. 10. We can observe that the content generated by GPT-4 in the zero-shot setting primarily follows the layout design rule, which further confirms the potential capability of LLMs in generating layouts when guided by the CIT approach. However, when compared to LayoutNUWA, there are several issues with the results generated by GPT-4: 1) the distribution of elements is uneven, with elements tending to be concentrated in certain areas, such as the left side of the canvas; 2) the element sizes are inconsistent, for instance, in some graphic layouts, there might be one or two large elements, which results in the high scores of the mIOU and Overlap metrics for some tasks; 3) there is a significant discrepancy between the data distribution of generated content and the real data.

| Task | Model | mIOU ($\downarrow$) | Align. ($\rightarrow$) | Overlap ($\rightarrow$) | FID ($\downarrow$) | Fail ($\downarrow$) |
|---|---|---|---|---|---|---|
| **Condition** $C \rightarrow S + P$ | GPT-4 (Zero-Shot) | **0.264** | 0.006 | 0.165 | - | 34.184 % |
| | LayoutNUWA-L2-DS | 0.260 | **0.021** | **2.898** | 9.741 | **0.000 %** |
| **Condition** $C + S \rightarrow P$ | GPT-4 (Zero-Shot) | 0.330 | 0.011 | **1.149** | - | 28.788 % |
| | LayoutNUWA-L2-DS | **0.358** | **0.020** | 2.483 | **4.682** | **0.000 %** |
| **Completion** | GPT-4 (Zero-Shot) | 0.362 | 0.044 | 0.728 | - | 28.535 % |
| | LayoutNUWA-L2-DS | **0.418** | **0.020** | **2.309** | 7.257 | **0.253 %** |
| **Real Data** | - | 0.348 | 0.016 | 1.521 | 6.695 | - |

Table 6: Detailed performance of GPT4 on the Magazine dataset. It is worth noting that due to the significant difference between the results generated by GPT-4 and the real data, the FID score cannot be calculated.

## C    HUMAN EVALUATION

We conduct the human evaluation for the model performance on the RICO and PubLayNet datasets. Specifically, We compare LayoutNUWA with two other strong baselines, including LayoutDM (In-oue et al., 2023) and LayoutTransformer (Yang et al., 2020), and randomly sample 25 graphic layouts generated from each model. We invite the annotators to choose which model performs better according to two evaluation settings: 1) *quality evaluation* based on the detail depiction, overlapping degree, and layout rationality in each layout; 2) *diversity evaluation* based on the diversity of the element arrangement in each layout. We hire 10 annotators to give their preferences, and the results are shown in Fig. 5(a) and Fig. 5(b). We can observe that layoutNUWA significantly outperforms the other two strong baselines, i.e., LayoutDM and LayoutTransformer, in terms of both generation quality and generation diversity. More generated cases can be referred to Fig. 10 (Magazine dataset) and Fig. 11 (RICO and PubLayNet datasets).

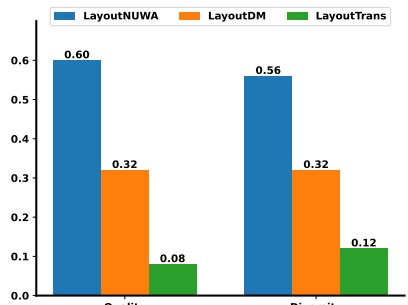
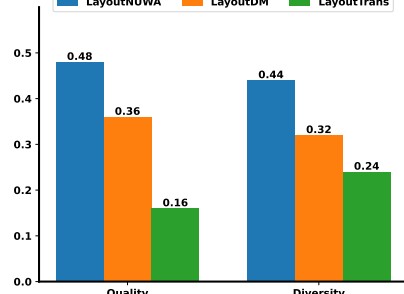

(a) Human evaluation on the RICO dataset.  (b) Human evaluation on the PubLayNet dataset.

---

[2]https://openai.com/policies/terms-of-use

## D  SAMPLED TRAINING-TESTING PAIRS WITH SIMILAR LAYOUT DISTRIBUTION

Considering that the qualitative generation results of LayoutNUWA look very similar to the actual design (Fig. 3), in order to further verify the fairness and rationality of our experiment, and to prove that the model is not overfitting on some specific training data, we used DocSim to select content from the training dataset that is similar to the Layout distribution of the test dataset. *We can observe that the layout distribution between the test data and some of the training data has some similarities. For example, in the PubLayNet dataset, the Title Element appears at the top of the overall layout, but it is not entirely consistent.*

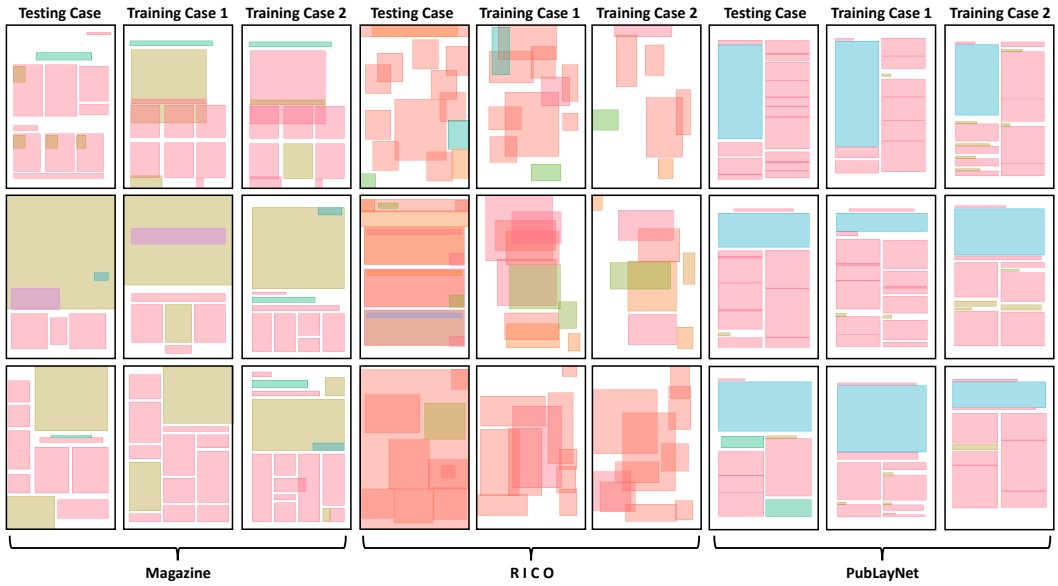

Figure 5: Sampled Training-Testing pairs by DocSim metric.

## E  MODEL PERFORMANCE UNDER UNCONDITIONAL SETTING

For unconditional generation, models generate 1,200 samples with the random seed. For LayoutNUWA-CL-DS, we randomly provide the empty templates without any prior information, e.g., category name, positions, *etc.*[3] However, since LayoutNUWA-CL-DA is trained under the domain-agnostic setting, we must provide it with the domain prompt, e.g., "I want to generate layout in mobile UI style" for the RICO dataset. We report Align. and FID score of generation results in Tab. 7, where we can observe that LayoutNUWA-CL-DS can obtain the best results on all FID scores and also achieve the best/comparable performance in Align. Score. As for the LayoutNUWA-CL-DA model, although the performance is not as good as the CL-DS setting, the results are still better than the baseline (some indicators are comparable to the baseline). We hypothesize that this may be due to potential domain confusion, i.e., the model has learned common features from previous datasets, which may have a negative effect on a specific domain. Stronger prior information is required to guide the model to generate layouts that are more consistent with a certain domain, such as element categories, *etc.*

For qualitative evaluation, we render the generation results in Fig. 9. We can observe that the graphic layouts generated by LayoutNUWA are more compact, with a lower failure rate, and their distribution is closer to that of real data. This can be attributed to the given template which defines the output format of the model and the utilization of the prior code knowledge from LLMs.

---

[3]This process is similar to the fixed length provided by the previous works.

| Models | RICO | | PubLayNet | | Magazine | |
|---|---|---|---|---|---|---|
| | Align. ($\rightarrow$) | FID ($\downarrow$) | Align. ($\rightarrow$) | FID ($\downarrow$) | Align. ($\rightarrow$) | FID ($\downarrow$) |
| LayoutTrans | 0.008 | 12.286 | 0.015 | 33.416 | 0.043 | 32.083 |
| MaskGIT | **0.003** | 60.724 | **0.004** | 28.836 | 0.003 | 157.560 |
| DiffusionLM | 0.019 | 23.997 | 0.016 | 18.720 | 0.043 | 37.951 |
| LayoutDM | 0.010 | 6.456 | 0.011 | 13.437 | 0.037 | 59.507 |
| LayoutNUWA-CL-DS (ours) | 0.008 | **5.674** | 0.007 | **8.914** | **0.092** | **24.108** |
| LayoutNUWA-CL-DA (ours) | 0.014 | 6.932 | 0.013 | 9.208 | 0.073 | 28.930 |
| **Real Data** | 0.004 | 6.250 | 0.001 | 1.850 | 1.693 | 6.695 |

Table 7: Model Performance on three testing sets under the **Unconditional Generation** setting.

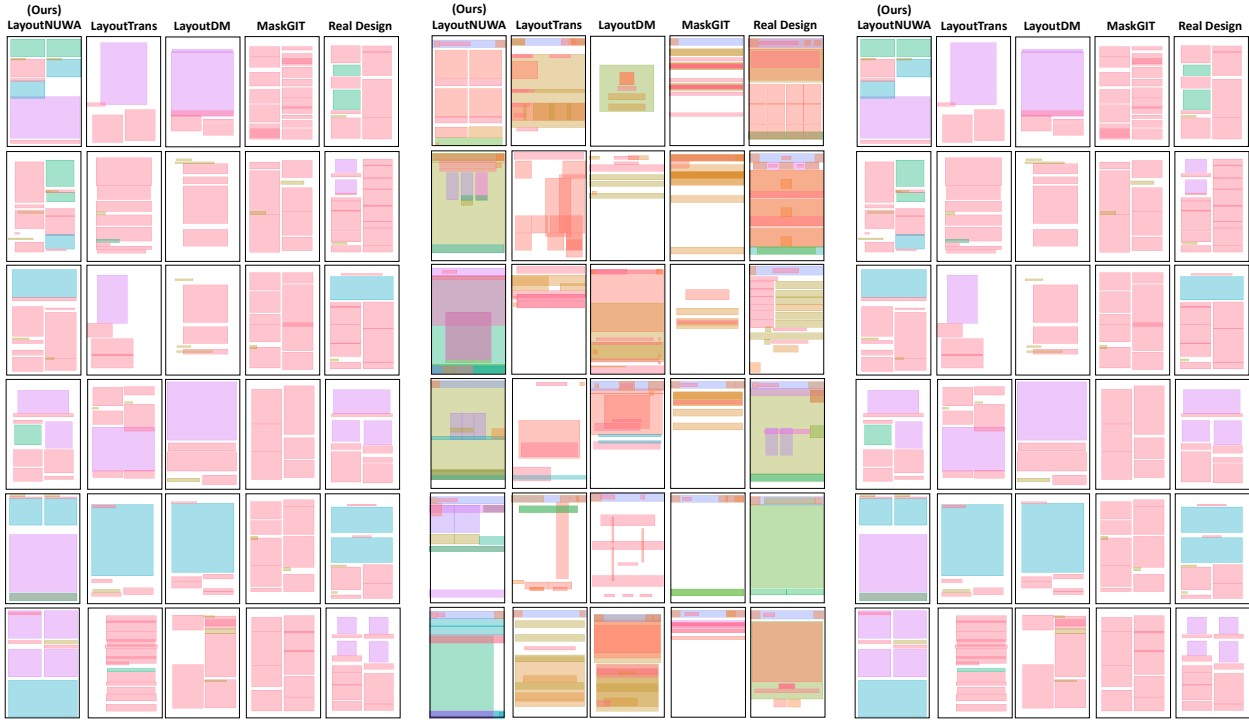

Figure 7: PubLayNet Dataset        Figure 8: RICO Dataset        Figure 9: Magazine Dataset

Figure 9: Cases generated by LayoutNUWA under the **Unconditional Generation** setting. We suggest zooming in on the monitor for better viewing.

## F    MODEL PERFORMANCE UNDER MIXED DOMAIN SETTING

Considering the impressiveness of the domain-agnostic model, we also take into account the issue of domain confusion (Long et al., 2022), that is, the data from one domain may affect another domain. Here, we design a toy experiment with the **mixed domain** setting, that is, applying the domain conditions of Magazine and RICO separately as prefixes as well as the code template behind using data from another domain. For example, we can utilize the below instruction to induce model generation under the Magazine $\rightarrow$ RICO mixed domain:

> *I want to generate layout in magazine style. Please generate the layout according to the {Task Condition} I provide: {RICO code template}*

As shown in Tab. 8, we can observe that when given more prior layout information, such as in the C + S $\rightarrow$ P setting, the model is less affected by domain confusion as the model tends to "resist" such

domain confusion, i.e., The performance of LayoutNUWA is much worse when domain confusion setting is applied compared to when it is not used. However, when there is a lack of layout prior conditions, such as in the C → S + P or Completion setting, the model is greatly affected by the domain confusion as such issue simply causes minor disturbances to the results, i.e., the model "yields to" the domain confusion under such circumstance. On one hand, this toy experiment reflects the significant impact of prior knowledge of code templates brought to the model, which can make LLM resist such domain confusion. On the other hand, the knowledge of the code template may have overwhelmed some of the information provided by the domain condition, which can lead to the generated content being weakly associated with the domain condition when there is insufficient code template information.

In future work, especially when adapting the LLMs to multiple layout distributions simultaneously, more attention needs to be paid to how to make models distinguish different domains by designing stronger constraints.

| Tasks | RICO → Magazine | | Magazine → RICO | |
|---|---|---|---|---|
| | Align. (→) | FID (↓) | Align. (→) | FID (↓) |
| **C + S → P** | - | - | 0.001 | 8.027 |
| w/o Domain-Confusion | **0.472** | **6.755** | **0.007** | **2.870** |
| **C → S + P** | 0.196 | 10.623 | 0.014 | 4.569 |
| w/o Domain-Confusion | **0.359** | **8.791** | **0.004** | **2.524** |
| **Completion** | 0.217 | 9.283 | 0.013 | 9.756 |
| w/o Domain-Confusion | **0.416** | **7.572** | **0.007** | **7.542** |
| **Real Data** | 1.693 | 6.695 | 0.004 | 6.250 |

Table 8: Model Performance under the **Mixed Domain** setting, where **A → B** indicates using A domain condition plus the B layout template. "-" denotes the failed generation.

## G  MORE TRAINING DETAILS

We deployed and trained the model based on the open-source LLM training framework LLaMA-X[4] and HuggingFace[5]. We optimized the training using DeepSpeed Zero3 technology (Rajbhandari et al., 2020). For the DS settings, we set the learning rate to 5e-5. For the DA settings, we set the learning rate to 5e-6 to prevent model explosion. We trained on 128 V100 units until the loss on the development set converged, at which point we stopped the training. We will open-source all the training details of this article after the anonymization period.

---

[4]https://github.com/AetherCortex/Llama-X
[5]https://huggingface.co/

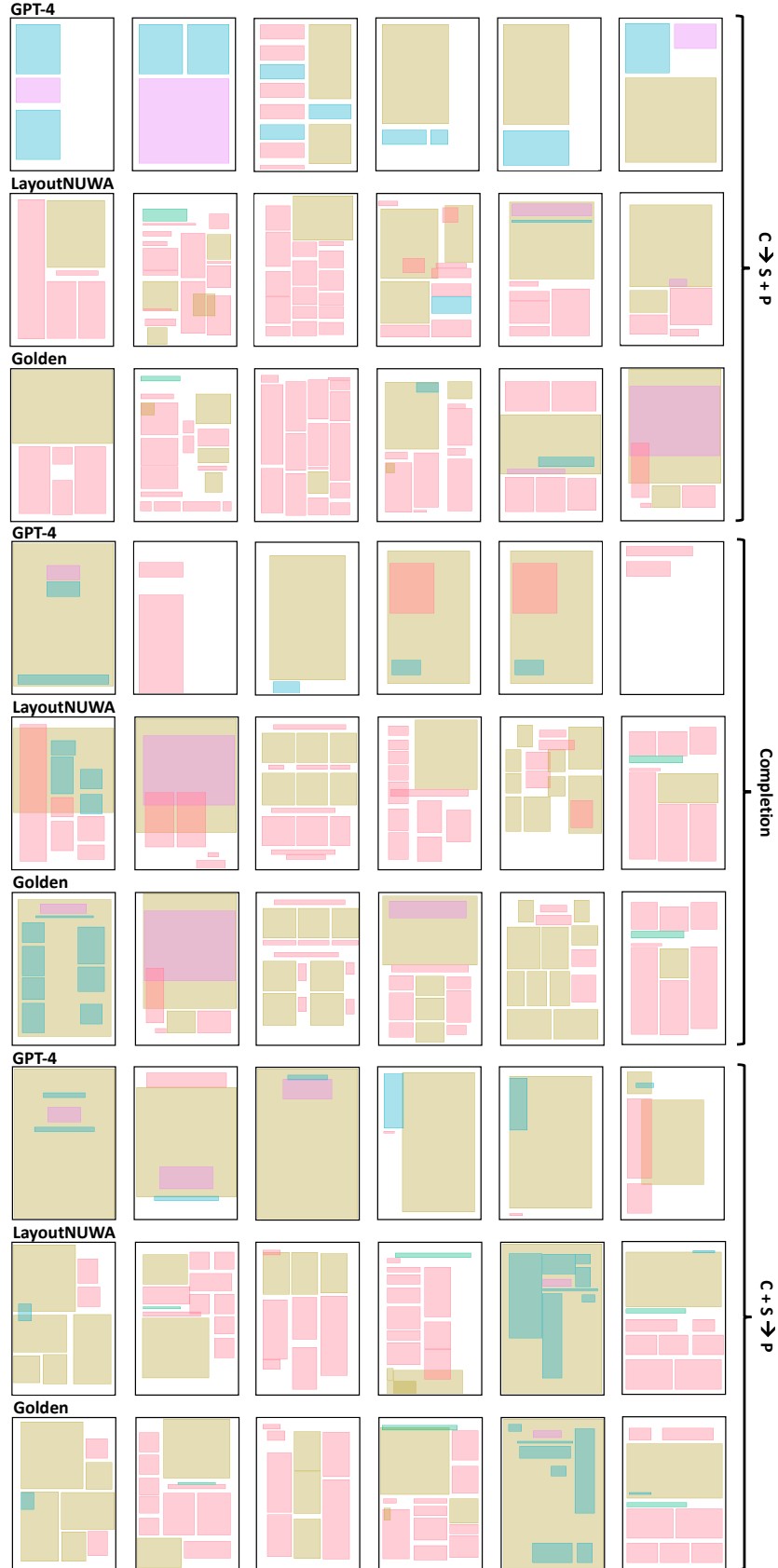

Figure 10: Comparison of rendered graphic layouts between GPT4 and LayoutNUWA on the Magazine dataset.

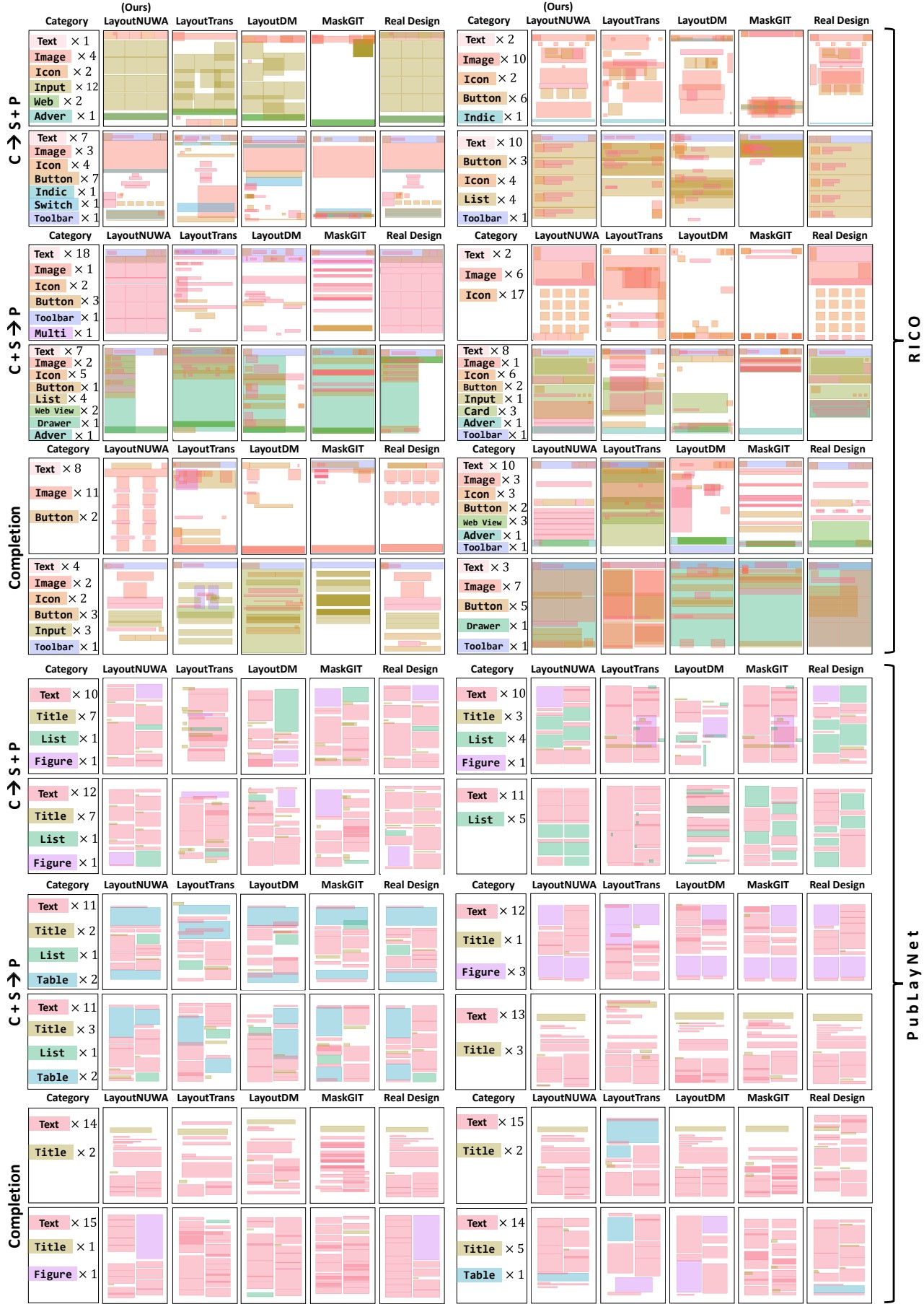

Figure 11: Cases generated by LayoutNUWA on the RICO and PubLayNet dataset.

