# OpenReview forum: "LayoutNUWA: Revealing the Hidden Layout Expertise of Large Language Models"
_ICLR.cc/2024/Conference — ICLR 2024 poster_

### Official Review · Reviewer_8sLn · 2023-10-31

**Soundness:** 1 poor
**Presentation:** 2 fair
**Contribution:** 3 good
**Rating:** 5
**Confidence:** 2

**Summary:**

The paper tackles conditional layout generation by formulating it as code generation and leveraging the prior knowledge of large language models (LLMs) to improve the performance of layout generation.

The main technical contribution is a code instruct tuning (CIT) method specialized in conditional layout generation, which consists of three components: 1) a code initialization module to convert input conditions into a masked HTML code; 2) a code completion module to predict the masked values using LLM; 3) a code rendering module to render the completed code into the final layout.

**Strengths:**

1. The idea of leveraging LLMs for conditions layout generation is very interesting, which has not been explored previously. The attempt of this paper on bridging LLMs and layout generation is inspiring, which provides a new perspective to layout modeling.

2. The performance improvement on the Magazine dataset is impressive.

**Weaknesses:**

1. The amount of technical contribution is limited. While the paper claims that a novel CTI approach is contributed, the technical innovation of the CTI is not significant since it only involves a simple solution to converting layout data to a LLMs-readable format (template generation), and some small modifications to adapt LLMs to layout generation (such as element order permutation and multi-task modeling). Therefore, the technical contribution that the paper can make to the ICLR community may not be significant.

2. The experiment setup is questionable. First, the proposed method, in theory, can support both conditional and unconditional layout generation (e.g., by setting the task instruction in Sec. 3.2.1 properly and tuning the LLM to generate the entire code instead of just the masked values). However, only the experiments on conditional generation are presented now, and there is no evaluation on unconditional generation that most of the previous works primarily aim for. Thus, a (potentially) key capability of the proposed method (also a problem interesting to the layout modeling community) is ignored in the paper. Second, LayoutGAN++ is only trained for the “C -> S + P” task. It is unclear how it is adapted to solve the other two conditional generation tasks in the experiments. Third, LayoutDM is only trained for unconditional generation, which is then adapted to conditional generation without any fine-tuning or re-training. In contrast, the proposed method is especially fine-tuned for various conditional generation tasks. As a result, direct comparison between LayoutDM and the proposed method under such a setting is unfair.

3. The performance on RICO and PubLayNet is unsatisfactory. As shown in Table 2, the advantage of the proposed method over SOTA methods is marginal on average. This is somewhat below expectation given the fact that the proposed method is based on large pre-trained models and has a chance to observe more training data (under the DA setting).

**Questions:**

1. In Eq.(3), what does M_j^t denote? What is the definition of L on the right hand side of the equation?

2. From the results in Table 1 and 2, the performance improvement on RICO and PubLayNet is not as significant as on the Magazine dataset. What are possible reasons behind this?

---

> ### Author Response · Authors · 2023-11-15
> **Response to Reviewer 8sLn**
>
> Firstly, thank you for your detailed suggestions and constructive comments.
>
> Regarding the issue you mentioned in **Weakness 2** (about the lack of unconditional generation results in the paper), we have provided a unified response in the General Response. Also, in the updated version of the paper, we have discussed in detail the effects of LayoutNUWA on unconditional generation in Appendix E (Page 16~17). Below is the experimental results:
>
> | Models | RICO Align. | RICO FID | PubLayNet Align. | PubLayNet FID | Magazine Align. | Magazine FID |
> |--------|-------------|----------|------------------|---------------|----------------|--------------|
> | LayoutTrans | 0.008 | 12.286 | 0.015 | 33.416 | 0.043 | 32.083 |
> | MaskGIT | **0.003** | 60.724 | **0.004** | 28.836 | 0.003 | 157.560 |
> | DiffusionLM | 0.019 | 23.997 | 0.016 | 18.720 | 0.043 | 37.951 |
> | LayoutDM | 0.010 | *6.456* | 0.011 | 13.437 | 0.037 | 59.507 |
> | LayoutNUWA-CL-DS (ours) | *0.008* | **5.674** | *0.007* | **8.914** | **0.092** | **24.108** |
> | LayoutNUWA-CL-DA (ours) | 0.014 | 6.932 | 0.013 | *9.208* | *0.073* | *28.930* |
> | Real Data | 0.004 | 6.250 | 0.001 | 1.850 | 1.693 | 6.695 |
>
> **Question 1:** About Equation Issue
>
> **Response 1:** In Equation 3, $M_{j}^{(t)} $represents the mask corresponding to the $j$-th element in a Layout for the $t$-th task.
>
> **Question 2:** About the performance of LayoutNUWA on different datasets.
>
> **Response 2:** This is a very astute observation. We will address your confusion from two aspects:
>
> 1. **The RICO and PublayNet datasets are large.** In previous works, even with not very large model parameters, the models perform well when trained on a large training dataset and then validated on a test set with a similar distribution. However, for datasets with less data, such as Magazine, the performance is poor. Therefore, our model improves more on the Magazine dataset. However, **we can't specifically look at the improvement on a certain dataset.** Specifically, we conducted Domain-Agnostic experimental settings, and LayoutNUWA has significant improvements on all datasets under this setting, which previous work could not achieve.
>
> 2. One of the success points of LayoutNUWA's good performance is that it uses the prior Code knowledge of LLM (Section 5.2, page 9). It's not surprising that it performs best on low-resource datasets (Magazine), which also proves our point that **LayoutNUWA makes full use of the prior code and layout knowledge of LLM.**

---

> > ### Comment · Reviewer_8sLn · 2023-11-23
> >
> > Thanks the authors for the responses. After reading them, I would like to maintain my current rating.

---

> > > ### Author Response · Authors · 2023-11-23
> > >
> > > Dear Reviewer 8sLn
> > >
> > > We are pleased to address your concerns and greatly appreciate your valuable comments!
> > >
> > > Best,
> > >
> > > Authors

---

### Official Review · Reviewer_qsQV · 2023-11-01

**Soundness:** 3 good
**Presentation:** 2 fair
**Contribution:** 2 fair
**Rating:** 6
**Confidence:** 5

**Summary:**

This paper introduces a new approach for the layout generation task. The proposed method treats layout design as an HTML code formulation and harnesses the power of a large language model to fill in the masked attributes. The proposed method operates in three primary phases: it begins by transforming the given conditions into an initialized HTML code. Subsequently, the language model is deployed to predict the final output including the masked attributes. Finally, this output is translated back into a layout design. The experiments on three datasets, (RICO, PubLay, and Magazine) show the improvement over the state-of-the-art methods in different settings.

**Strengths:**

The paper presents is easy to follow. The proposed approach to utilizing the capabilities of LLMs for layout design, particularly through the formulation of HTML codes, is novel to the best of my knowledge. Moreover, the proposed technique surpasses current state-of-the-art methods, especially when tested on the Magazine dataset.

**Weaknesses:**

A significant limitations that in my view is its inability to incorporate non-categorical data, such as image embeddings. This constraint considerably narrows the method's applicability in real-world situations, especially when users input attributes that aren't readily translatable for language models.

The presentation could benefit from further refinement. Specific implementation details are missing, including the number of training iterations, batch size, learning rate, etc. Furthermore, Figure 3's organization makes it challenging to visually compare the proposed method against both the original design and LayoutDM as the state-of-the-art method.

Given that prior techniques were optimized for domain-specific settings, the enhancements seen in the "LayoutNUWA-*-DS" variants don't come across as particularly robust, as evidenced by metrics like the FID in the RICO dataset. This limitation, coupled with the inherently slower processing speed of LLMs compared to other methods, may limit the method's broader application.

**Questions:**

As I understand, the authors also pass the layout size as the initialization part. However, this is not given in for example LayoutDM to the best of my knowledge and can be a strong signal to the model. Can you clarify on this part?

---

> ### Author Response · Authors · 2023-11-15
> **Response to Reviewer qsQV**
>
> Thanks for your patient review.
>
> **Question 1: about the issue of input condition containing layout size.**
>
> **Response 1:** That’s a good question. In previous papers (including LayoutDM), *a specific template containing the layout size information is provided during the rendering process.* During the training process of the model, although the layout size information is not integrated, some restrictions are applied to the data, such as encoding relative position/size information (in decimal form) instead of absolute position/size encoding, which also *indirectly incorporates the predetermined layout size information.*
>
> For LayoutNUWA, we believe that providing layout size information has several benefits:
>
> 1. The results generated by the model can be directly rendered, so layout size information is required;
>
> 2. We use absolute position/size encoding, which is closer to the real scenario and utilizes the code prior knowledge of LLM. Compared with relative position encoding, the model's generated results are more interpretable, so a layout size is needed as a prior condition to constrain absolute position/size encoding, which *actually makes higher demands on the model's generation, i.e., modeling with absolute value is much harder than the relative value.*
>
> 3. Layout size also indirectly reflects domain characteristics. For example, the RICO dataset is a mobile UI layout, which requires typesetting in a tighter space (so there will be some overlap), while the Magazine dataset has a larger space, so typesetting has less overlapping. This is especially important for LLM to do domain-agnostic (DA) tasks.
>
> **Question 2:** About the lack of training details.
>
> **Response 2:** This is a great suggestion. We provide more details here:  we deployed and trained the model based on the open-sourced LLM training framework [LLaMA-X](https://github.com/AetherCortex/Llama-X) and [HuggingFace](https://huggingface.co/). We optimized the training using DeepSpeed Zero3 technology. For the DS settings, we set the learning rate to 5e-5. For the DA settings, we set the learning rate to 5e-6 to prevent model explosion. We trained on 128 V100 units until the loss on the development set converged, at which point we stopped the training.
>
> **Note:** For more details about training and deployment, please refer to the latest version of the paper (**Appendix G, page 18**).

---

### Official Review · Reviewer_dDf9 · 2023-11-01

**Soundness:** 3 good
**Presentation:** 3 good
**Contribution:** 2 fair
**Rating:** 6
**Confidence:** 3

**Summary:**

This paper tackles the task of graphic layout generation. Previous works usually treat this task as a numerical optimization problem, whereas this paper proposes to leverage the knowledge of LLMs for layout generation. Specifically, the authors propose a Code Instruct Tuning framework with three components: Code Initialization, Code Completion, and Code Rendering. The proposed method achieves strong performance compared with current SOTAs on three datasets, RICO, PubLayNet, and Magazine.

**Strengths:**

- The paper is well organized and easy to read.

- Utilizing LLMs for layout generation seems a good way to tackle this graphic layout generation task. The code template also supports three different variants for code completion task.

- The paper provides strong results compared with baseline models, as well as several ablation studies including LLaMA2/CodeLLaMA, Domain-Specific/Domain-Agnostic settings, and code/numerical output formats.

**Weaknesses:**

- This paper focus on graphic layout generation. Even though LLMs are not used for this specific task, layout generation via LLMs seems not novel [1,2,3]. Therefore, the novelty of this paper is not very strong.



[1] Cho, Jaemin, Abhay Zala, and Mohit Bansal. "Visual Programming for Text-to-Image Generation and Evaluation." arXiv preprint arXiv:2305.15328 (2023)

[2] Feng, Weixi, et al. "LayoutGPT: Compositional Visual Planning and Generation with Large Language Models." arXiv preprint arXiv:2305.15393 (2023)

[3] Lian, Long, et al. "LLM-grounded Diffusion: Enhancing Prompt Understanding of Text-to-Image Diffusion Models with Large Language Models." arXiv preprint arXiv:2305.13655 (2023).

**Questions:**

The authors mention that LayoutNUWA suffers from the error propagation problem. Some visualizations/examples might be useful to illustrate this point.

**Details Of Ethics Concerns:**

No ethics concerns.

---

> ### Author Response · Authors · 2023-11-15
> **Response to Reviewer dDf9**
>
> Thanks for your valuable comments.
>
> **Question 1:** About the issue of understanding cases of error propagation.
>
> **Response 1:** We have updated the example diagram of error propagation (Appendix A, Figure 4, page 14) in the updated version of the PDF.  You can also click on this link: [https://anonymous.4open.science/r/LayoutNUWA-ICLR2024-Anonymous-Supplementary-Materials/error_propagation.pdf](https://anonymous.4open.science/r/LayoutNUWA-ICLR2024-Anonymous-Supplementary-Materials/error_propagation.pdf) to view it on the anonymous website. *( If a PDF rendering error occurs, manually refreshing the page can resolve it )*
>
> Specifically, possible error propagation is that the history layout generated by the AR model contains an unreasonable Layout, which will affect the layout generation of subsequent elements, such as overlapping, unreasonable size/location, etc. However, these all happen under the condition of insufficient training. **All models in this paper have been fully trained, so the impact of this error propagation is minimal.**

---

### Official Review · Reviewer_7ZDJ · 2023-11-02

**Soundness:** 3 good
**Presentation:** 4 excellent
**Contribution:** 3 good
**Rating:** 8
**Confidence:** 3

**Summary:**

This paper introduces a novel method to instruction-tune LLMs for layout generation task. Utilizing pre-trained LLMs and formulated as a code-generation task, the proposed method can build upon the semantic knowledge about geometries and code in the LLMs and achieve SoTA performance for conditional layout generation tasks. This paper further explores both domain-specific and domain-agnostic settings, demonstrating that a single LLM instruction-tuned in multiple domains can achieve SoTA performance in multiple domains.

**Strengths:**

- Achieved SoTA in Layout Generation tasks in multiple domains with a single LLM.
- Extensive experimentation with multiple baselines, model variants, tasks and datasets.
- Enabled LLMs to generate layouts, which opens up possible research avenues for complex language-based interactions in the layout generation space.

**Weaknesses:**

- *Limited Architectural Novelty from Fine-tuning LLMs*
- *Constrained Tasks and Conditions*: While the evaluated tasks are established and important for fairly comparing against existing work, I believe the paper can be significantly improved by exploring more complex, language-based interactions, given that LLMs are used to perform the layout generation task for the first time. It would significantly improve the contribution if the author(s) can explore (could be preliminary) whether LLMs can fuse its natural language / common task knowledge with layout generation, such as having a conversation with users about their layout generation constraints, or specifying the use-cases of the layouts.
- *Potential limited diversity and overfitting*: The qualitative generation results appear very similar to the real design, and it is unclear whether the model can generate diverse layouts given a single condition, or if the model has overfitted to a certain specific layout. I suggest the author(s) to include the most similar training example to the generation (by DocSim) in a figure in the appendix. Please also see questions below for addressing my concerns.

Because of these concerns, I am currently on the fence about accepting this paper, but I am willing to raise my score if these concerns are addressed adequately in the rebuttals.

**Questions:**

- Regarding the potential diversity and overfitting, I wonder if the author(s) can show whether the model can generate diverse layouts during the sampling process, given the same input conditions. Moreover, I wonder if the 'real design' in the Figures (e.g., Fig 5) are taken from the training or validation set? It could indicate overfitting given the high similarity between real designs and the generated designs in LayoutNUWA, especially given the LLMs  are significantly larger in size over prior work.

- Have the author(s) also explored unconditional generation? It would be good to understand the capability of the models to generate diverse layouts unconditionally, following the point above.

- While the domain-agnostic model is impressive, I wonder if the author(s) have observed domain confusion? Such as the model generating a magazine layout (when prompted to generate a mobile layout (from RICO).

---

> ### Author Response · Authors · 2023-11-15
> **Response to Reviewer 7ZDJ**
>
> Thank you for your constructive comments.
>
> **Question 1:** About generation diversity and overfitting.
>
> **Response 1:** This is a great question. We construct the same input and take the top 4 paths with the highest probabilities from the model output. Our model can generate diverse layouts in the position and size of each element while keeping the overall position of elements unchanged. You can see the generated cases on our provided anonymous website in  [https://anonymous.4open.science/r/LayoutNUWA-ICLR2024-Anonymous-Supplementary-Materials/README.md](https://anonymous.4open.science/r/LayoutNUWA-ICLR2024-Anonymous-Supplementary-Materials/README.md), and this phenomenon is quite obvious. In the paper, all real designs are sampled from the test and development datasets (which is completely consistent with previous work such as LayoutDM). To further address your concerns, in the updated version of the paper, we used the DocSim metric in Appendix D (page 16) to show the distribution of training data similar to the test set, which also proves that our model is not overfitting on some specific layout distributions.
>
> **Question 2:** About unconditional generation.
>
> **Response 2:** That's a great suggestion. In the updated paper, we also explored the unconditional generation in Appendix E (page 16~17). Here is the model performance:
> | Models | RICO Align. | RICO FID | PubLayNet Align. | PubLayNet FID | Magazine Align. | Magazine FID |
> |--------|-------------|----------|------------------|---------------|----------------|--------------|
> | LayoutTrans | 0.008 | 12.286 | 0.015 | 33.416 | 0.043 | 32.083 |
> | MaskGIT | **0.003** | 60.724 | **0.004** | 28.836 | 0.003 | 157.560 |
> | DiffusionLM | 0.019 | 23.997 | 0.016 | 18.720 | 0.043 | 37.951 |
> | LayoutDM | 0.010 | *6.456* | 0.011 | 13.437 | 0.037 | 59.507 |
> | LayoutNUWA-CL-DS (ours) | *0.008* | **5.674** | *0.007* | **8.914** | **0.092** | **24.108** |
> | LayoutNUWA-CL-DA (ours) | 0.014 | 6.932 | 0.013 | *9.208* | *0.073* | *28.930* |
> | Real Data | 0.004 | 6.250 | 0.001 | 1.850 | 1.693 | 6.695 |
>
> **Question 3:** About the domain confusion issue.
>
> **Response 3:** This is a great question, and we specifically discussed this issue in Appendix F (page17~18). As shown in the table below:
>
> | Tasks | RICO -> Magazine Align. | RICO -> Magazine FID | Magazine -> RICO Align. | Magazine -> RICO FID |
> |--------|------------------------|----------------------|------------------------|----------------------|
> | C + S -> P | - | - | 0.001 | 8.027 |
> | w/o Domain-Confusion | **0.472** | **6.755** | **0.007** | **2.870** |
> | C -> S + P | 0.196 | 10.623 | 0.014 | 4.569 |
> | w/o Domain-Confusion | **0.359** | **8.791** | **0.004** | **2.524** |
> | Completion | 0.217 | 9.283 | 0.013 | 9.756 |
> | w/o Domain-Confusion | **0.416** | **7.572** | **0.007** | **7.542** |
> | Real Data | 1.693 | 6.695 | 0.004 | 6.250 |
>
> , where we design a toy experiment and find: 1)when given more prior layout information, such as in the C+ S → P setting, the model is less affected by domain confusion as the model tends to “resist” such domain confusion, i.e., The performance of LayoutNUWA is much worse when domain confusion setting is applied compared to when it is not used; 2) However, when there is a lack of layout prior conditions, such as in the C → S + P or Completion setting, the model is greatly affected by the domain confusion as such issue simply causes minor disturbances to the results, i.e., the model “yields to” the domain confusion under such circumstance. Overall, under the DA setting, the model indeed experiences Domain Confusion problems (mainly depending on the amount of given prior knowledge), which is enlightening when using LLM for DA Layout modeling. For more details, please refer to Appendix F of this updated paper.

---

> ### Comment · Reviewer_7ZDJ · 2023-11-22
>
> Thank you for addressing my concerns. I have raised my score from 6 to 8.

---

> > ### Author Response · Authors · 2023-11-22
> >
> > Dear Reviewer 7ZDJ
> >
> > Thank you for your support! We are pleased to address your concerns and greatly appreciate your valuable comments, which play a crucial role in improving our work.
> >
> > Best,
> >
> > Authors

---

### Author Response · Authors · 2023-11-15
**General Response to All Reviewers**

Firstly, I'd like to express gratitude to all the reviewers for their constructive suggestions and patient review of this paper.

**We need to emphasize that we have updated the submitted paper. Reviewers can refer to our latest uploaded paper and our response for checking.**

Considering some common issues, we will provide a common response list.
1. We note that both Reviewer 7ZDJ and 8sLN both mentioned the lack of an unconditional generation experiment in the paper. This is a good point, and in our updated paper, we have supplemented it with the results of the unconditional generation experiment as well as some rendering cases (Appendix E & F, Table 7 & 8, Figure 7). Below, we also provide the results of unconditional generation.

| Models | RICO Align. | RICO FID | PubLayNet Align. | PubLayNet FID | Magazine Align. | Magazine FID |
|--------|-------------|----------|------------------|---------------|----------------|--------------|
| LayoutTrans | 0.008 | 12.286 | 0.015 | 33.416 | 0.043 | 32.083 |
| MaskGIT | **0.003** | 60.724 | **0.004** | 28.836 | 0.003 | 157.560 |
| DiffusionLM | 0.019 | 23.997 | 0.016 | 18.720 | 0.043 | 37.951 |
| LayoutDM | 0.010 | *6.456* | 0.011 | 13.437 | 0.037 | 59.507 |
| LayoutNUWA-CL-DS (ours) | *0.008* | **5.674** | *0.007* | **8.914** | **0.092** | **24.108** |
| LayoutNUWA-CL-DA (ours) | 0.014 | 6.932 | 0.013 | *9.208* | *0.073* | *28.930* |
| Real Data | 0.004 | 6.250 | 0.001 | 1.850 | 1.693 | 6.695 |


2. In addition, we have provided more generated cases and some additional conclusions, which may give you a better understanding of the effects of LayoutNUWA and this method of modeling Layout using LLM and Code, which have been posted on the anonymous website: [https://anonymous.4open.science/r/LayoutNUWA-ICLR2024-Anonymous-Supplementary-Materials/README.md](https://anonymous.4open.science/r/LayoutNUWA-ICLR2024-Anonymous-Supplementary-Materials/README.md)

3. We have updated more training details in this uploaded PDF (Appendix G, page 18) and commit to releasing our code after the anonymization period, to help the Layout open-source community better explore and utilize the paradigm of generating Layout with LLMs.

---

> ### Author Response · Authors · 2023-11-20
> **Looking forward to further discussion**
>
> Dear Reviewers,
>
> Thanks for your review. Considering the response deadline is approaching, we hope to have more discussions with you.
> The corresponding responses are already presented in each review's response box. In addition to this:
>
> 1. We have added **more experimental results** and **updated the submitted PDF**;
> 2. We have provided an **anonymous website** [https://anonymous.4open.science/r/LayoutNUWA-ICLR2024-Anonymous-Supplementary-Materials/README.md](https://anonymous.4open.science/r/LayoutNUWA-ICLR2024-Anonymous-Supplementary-Materials/README.md) showing more experimental results.
>
> **If you have any further inquiries or suggestions, please do not hesitate to reach out to us.**

---

### Meta-Review · Area_Chair_5fg8 · 2023-12-08

**Metareview:**

The paper presents LayoutNUWA, which investigates LLMs for layout generation. The basic idea is to express generation conditions as natural language and markup languages that is expected by LLMs, with spatial information represented as mask tokens. The rest of the process is just to ask LLMs to fill the mask like code completion. The approach doesn't seem to require much post processing. The overall approach is simple and involves instruction tuning of LLMs.

All the reviewers like the idea of using LLMs for layout generation. Reviewer dDf9 noted that "Utilizing LLMs for layout generation seems a good way to tackle this graphic layout generation task. " and Reviewer 8sLn "The idea of leveraging LLMs for conditions layout generation is very interesting, which has not been explored previously. ". There are actually many attempts out there using LLMs for layout generation. But this work shows a more systematic investigation for LLMs's ability for layout generation, rather than just anecdotally showing successful examples. The reviewers also commended "Extensive experimentation with multiple baselines, model variants, tasks and datasets."Reviewer 7ZDJ.

That said, the technical novelty of the work is low as pointed out by all the reviewers. Particularly, Reviewer 8sLn pointed out that "The amount of technical contribution is limited. While the paper claims that a novel CTI approach is contributed, the technical innovation of the CTI is not significant since it only involves a simple solution to converting layout data to a LLMs-readable format (template generation), and some small modifications to adapt LLMs to layout generation (such as element order permutation and multi-task modeling). Therefore, the technical contribution that the paper can make to the ICLR community may not be significant." It is the case that in the past papers have explored approaches of using HTML-like representation for LLM-friendly input. Even the most positive reviewer, Reviewer 7ZDJ, criticized the novelty of the work: "Limited Architectural Novelty from Fine-tuning LLMs". The reviewers also pointed out a number of issues with the paper regarding diversity and overfitting, which have been constantly observed when fine-tuning LLMs for this purpose.

In sum, the paper reports a set of interesting findings, yet it doesn't seem to make enough contribution for technical novelty and depth.

SAC comment: Since this work does a good systematic investigation, that brings novelty to our knowledge without algorithmic innovation. I am inclined to accept.

**Justification For Why Not Higher Score:**

The technical novelty and depth of the work are limited.

**Justification For Why Not Lower Score:**

The paper explores an exciting topic of using LLMs for layout generation.

---

### Decision · Program_Chairs · 2024-01-16

Accept (poster)